# Garrano Horses Perceive Letters of the Alphabet on a Touchscreen System: A Pilot Study

**DOI:** 10.3390/ani12243514

**Published:** 2022-12-12

**Authors:** Clara-Lynn Schubert, Barbara Ryckewaert, Carlos Pereira, Tetsuro Matsuzawa

**Affiliations:** 1Department of Cognitive and Behavioral Neurosciences, Faculty of Science and Engineering of Sorbonne University, CEDEX 05, 75005 Paris, France; 2Laboratory «Scenes of the World, Creation, Critical Knowledge», Doctoral School of Aesthetics, Sciences and Technology of the Arts, University of Paris 8 Vincennes, CEDEX 02, 93526 Saint-Denis, France; 3Department of LEA (Applied Foreign Languages), University of Paris 3 Sorbonne Nouvelle, CEDEX 05, 75230 Paris, France; 4Division of Humanities and Social Sciences, California Institute of Technology, Pasadena, CA 91125, USA; 5Department of Pedagogy, Chubu Gakuin University, Gifu 504-8037, Japan; 6College of Life Sciences, Northwest University, Xi’an 710069, China

**Keywords:** horse cognition, letters of the alphabet, shape perception, touchscreen system, operant conditioning: shaping, one-male unit, animal welfare

## Abstract

**Simple Summary:**

The present study used a touchscreen system to study the visual perception of Garrano horses, an endangered breed of pony belonging to the Iberian horse family. The participant horses (13, 8, 5, 4, and 2 years old) were kept as a one-male unit (OMU), living together permanently in an enriched environment near their natural habitat in Serra d’Arga, northern Portugal. Through successive training stages, all five horses acquired the skill of nose touch. All horses except the male learned to discriminate five letters of the alphabet, namely O, B, Z, V, and X. The error patterns and the analysis of shape features showed that the curved letters, O and B, look similar to the horses, as do the straight-line letters, Z, V, and X. The result is congruent with previous studies of shape perception in other animals. Because of the nature of the automated system, the touchscreen discrimination was free from social cueing known as the “Clever Hans phenomenon”. Without a lead rope, the horse participated in the visual cognition task presented by the touchscreen system. This is a pilot study for horses to freely start and engage the cognitive tests and to freely stop and return to the social group.

**Abstract:**

This study aimed to use a computer-controlled touchscreen system to examine visual discrimination in Garrano horses (*Equus caballus*), an endangered breed of pony belonging to the Iberian horse family. This pilot study focused on the perceptual similarity among letters of the alphabet. We tested five horses in a one-male unit (OMU) living permanently in a semi-free enclosure near their natural habitat in Serra d’Arga, northern Portugal. Horses were trained to nose-touch black circles that appeared on the screen. Then, they were tested for discrimination of five letters of the Latin alphabet in Arial font, namely O, B, V, Z, and X, using a two-choice discrimination task. The confusion matrix of letter pairs was used to show the MDS and to identify the relative contribution of shape features. The results showed perceptual similarities among letters with curvatures pitted against those of straight lines. Shape perception in horses seems to share features with that of humans and other animals living in different niches. The touchscreen system proved to be an objective and innovative way of studying cognition in the socially organized group of horses. The automated system can promote the welfare of captive horses by maximizing their freedom of movement.

## 1. Introduction

Humans and horses have had a long-term relationship spanning over 5500 years [1]. Today, horses are very popular in sports, leisure, and other activities. A parallel approach consisting of fieldwork and laboratory work may help to better understand horses (*Equus caballus*) from a holistic perspective [2]. Fieldwork was applied to feral horses such as the Garranos in northern Portugal, an endangered breed of pony belonging to the Iberian horse family [3,4]. Recent advances in field studies utilized drones and have clarified the social structure and social life of Garranos [5,6,7,8,9,10,11,12]. They are group-living animals with a multi-level society (MLS), similar to human societies and some nonhuman societies [13,14]. The aggregation of feral horses is usually called a “herd,” which consists of multiple one-male units (OMUs) and all-male units called bachelor groups. The OMU consists of one adult male (“stallion”) and multiple females and their dependents. In short, the MLS consists of the double layers of the herd and its units. Social contact among individual horses in the OMU is essential for keeping the society [7,8,12]. Even more, Garrano horses in Serra d’Arga are predated by wolves [15,16,17]. Iberian wolves (*Canis lupus signatus*) affect these free-ranging feral Garranos. The Garrano breed might be an interesting subject to study how horse cognition has been shaped through society and ecology. However, in contrast to field studies, there are no previous studies on the cognition and behavior of Garrano horses. This pilot study aimed to set up a cognitive study of Garranos in a laboratory near their natural habitats, Viana do Castelo district near Serra d’Arga, northern Portugal.

Many excellent laboratory studies have been conducted on horse cognition for horses other than Garranos [18,19,20,21]. See the recent review by Brubaker and Udell [22]. For example, there have been studies of animal models of depression [23], emotion [24], sensory sensitivity and temperament [25], cross-modal representation [26,27], auditory and other sensory processing [28], exploratory behavior [29], and social cognition [30,31,32,33]. Focusing on visual cognition, the topics cover face recognition [34], long-term memory [35], visual laterality and possible hemispheric asymmetry [36,37], and interocular transfer [38]. However, very little is known about the fundamental visual perception of horses [39]. We know about color vision [40] and visual acuity [41]. Horses are grazing herbivores who have adapted to terrestrial environments by developing a very wide visual field, reflected in the lateral location of their eyes. The anatomical characteristics may have affected their visual perception. Few studies have investigated fundamental visual perception in horses, and many issues remain to be addressed such as categorical perception, depth perception, motion perception, and shape discrimination.

The main topic of the present study was shape discrimination in horses, particularly shapes with curves and shapes with straight lines such as diagonal (oblique) and/or horizontal/vertical lines in discriminating letters of the Roman alphabet. Letters of the alphabet have been used in the study of human information processing [42] and comparative cognition studies among different species, including chimpanzees [43], pigeons [44,45], dolphins [46], and horses [47]. Letters of the alphabet are ubiquitous in human society. The letters provide the basis of human visual communication. Horses in laboratories are known to do symbolic communication with human caretakers [48]. Therefore, letters of the alphabet learned by horses can be used for communication purposes just like chimpanzees [43]. The perception of letters can be the first step to comparing visual information processing directly with humans and may also open the way to zoo semiotics in horses [49]. Moreover, knowing how horses perceive their environment will enable us to improve their welfare in various situations.

Unfortunately, many previous perceptual and cognitive studies on horses were affected by social cueing—humans were heavily involved in the testing procedures. There is a long history of debate about the so-called “Clever Hans phenomenon”, where a horse responded directly to involuntary cues in the body language of the trainer [50]. To eliminate the possibility of social cueing, Tomonaga et al. (2015) first applied a computer-controlled touchscreen system to evaluate shape discrimination in horses [47]. The automated touchscreen method was inspired by the experimental paradigm established for the long-term study of perception, cognition, and memory in chimpanzees [51,52].

The present study followed the original touchscreen study by Tomonaga et al. (2015) to further explore the visual perception of five Garrano horses. The strategy was to avoid experimental bias from social cueing as much as possible. The “touchscreen” combines an input (called “touch panel”) and an output device (called “display” or “monitor”). In terms of behavioral control and operant conditioning, this stimulus-response spatial contiguity helps the horses in discrimination tasks. Thus, the computer-controlled touchscreen system might be the best currently available method and was therefore employed for this experiment.

In addition to the touchscreen, this study aimed to establish a new way of testing horses in captivity. The key point is setting up a socially organized OMU, simulating feral horses’ natural behavior. The horses studied here live semi-free in a large enclosure near their natural habitat; they are not separated in stables. As the horses were allowed to move freely in the testing area, they could choose to engage or not engage in the task. The touchscreen system can maximize the freedom of group-living horses while studying their perceptual abilities. There is another related issue which concerns the welfare of disabled animals. One of the five horse participants in the present study is half-blind, due to a damaged right eye. Deprived of binocular cues, she is limited to completely lateralized monocular cues. As we wanted to find a way to include this horse in the present study, the feasibility of using the touchscreen system for the disabled horse was also explored.

## 2. Materials and Methods

### 2.1. Participants

**Horses:** We studied five ponies (*Equus caballus*) called “Garranos” in northern Portugal [3,4]. A pony is defined as a mature horse below a certain height at the withers; this may vary from about 142 cm h to nearly 150 cm h. The horses were kept together as a group in an outside enclosure built for horse studies in the town of Lanheses, Viana do Castelo, northern Portugal. It is near the natural habitat of feral Garrano horses in Serra d’Arga (Figure 1).

The five horses were between 2 and 13 years old (see Table 1, Appendix A). The females but not the male were kin-related. At two years old, the youngest horse (Kiki) still needed maternal social care. As she was not used to being separated from her mother, the two horses were initially trained together. The male (Boneko) was castrated but behaved like a stallion in a feral group. Thus, the group of five subjects approximated the natural social grouping: a one-male unit (OMU). The half-blind horse, Flore, lost sight in her right eye as a result of an accident when she was 5 years old. All five horses were born in captivity. Only Flore and Boneko had taken part in horse-riding activities; the others were less used to human contact. They were all naïve in terms of perceptual-cognitive experiments.

To check the vision status of each horse, the menace reflex was tested. Closure of the horse’s eyelids was expected when the experimenter moved his hand quickly towards one of the two eyes and stopped it abruptly at around 15 cm from the eye. This menace reflex was present in both eyes for Boneko, Petala, Noven, and Kiki, but only in the left eye for Flore. Although Flore’s right eye did not show the menace reflex, she does appear sensitive to the presence of a shadow on her right side: she sometimes turned her head when a veterinarian placed himself in front of her right eye. For the OMU, Flore was essential as a mare who had two offspring in the social group although she is disabled in one eye. According to previous studies, the left eye of horses is important to identify humans [36,37]. The left eye of Flore was intact.

**Care:** A team of people shared the care and feeding of the horses. The horses were fed three times a day and were not food-deprived for the experiments. They had free access to water. They were fed with natural grassland hay between 2 and 3 times per day, distributed in 3 different large wooden boxes. They were also fed wet wheat bran twice a day, mornings and evenings, completed by a handful of mixed cereals.

The horses were not isolated in each stable. The social group was permanently free on the outside. The housing pen with the shelter for the touchscreen experiment was 3500 m^2^. The housing rotates with another field of 9000 m^2^. This rotation maintains natural grassland for horses and is organized at the beginning of fall and spring.

Regular human interaction was not present. It mainly consisted of basic manipulations such as grooming, walking in hand, lunging, and riding lessons. Riding lessons only concerned Boneko and Flore. The horses had fewer interactions in the winter because of the weather. However, in the spring and summer, they had more interactions and more grooming. The horse riding was stopped during the study period. All tests were noninvasive and followed the ethical guidelines of the Garrano Horse Center (GHC) where the participant horses live and the study was carried out. The study received the appropriate ethical clearance (see Institutional Review Board Statement).

### 2.2. Apparatus

**Touchscreen and computer:** A touchscreen (IIYAMA Prolite T2250MTS touchscreen monitor—High Definition 1920 × 1080 pixels, resolution with a 5-msec response time, Tokyo, Japan) was used to show the stimuli and detect the nose touch by the horses. It measured 43” (98 cm × 58 cm) and was set up on a portable stand (94 cm × 178 cm) holding the apparatus. The center of the touchscreen was located 123 cm above ground level. This was slightly lower than the average wither height of the five horses to facilitate an adequate nose touch. A computer (Panasonic Toughbook CF-C2) controlled the screen monitor and was placed behind it on a small table, out of the horse’s view (see Appendix A).

The test setup consisted of the touchscreen and the feeder (food container). It remained inside a square stall (280 cm × 280 cm) equipped with a video camera. The horse was prevented from entering the stall by a 70-cm high wooden barrier, enabling the horse to put his/her neck through the stall opening in front of the screen (Figure 2).

**Food reward and dispenser:** Carrot pieces were used as food rewards during the experiment. They were small cubes about 2 × 2 × 2 cm, weighing 2 g on average. The reward was delivered manually through a tube (3.5 cm in diameter and 70 cm long). It landed in a bowl (22 cm diameter) located below the touch panel in front of the horse. The top opening of the tube was set behind the screen so the horse could not see the experimenter dropping the carrot cube into the tube. The inter-trial interval was fixed at 5 sec. The horse received the reward carrot through the dispenser, not directly from the experimenter in the discrimination task. When the session ended, an apple slice was given manually by the experimenter before bringing the pony back to the group of ponies. It involved more interactions with the horse after the discrimination task.

**Programming:** The experiment was programmed through two different online platforms: PsychoPy^®^ and Pavlovia^®^. PsychoPy^®^ (https://www.psychopy.org/ accessed on 13 July 2022) is a free cross-platform package that runs a wide range of experiments in the behavioral sciences; it programmed the present experiments. Pavlovia^®^ (https://pavlovia.org/ accessed on 13 July 2022) was originally conceived as a repository and launch platform for PsychoPy^®^ experiments; it stored the data from each session.

### 2.3. Stimuli

**Black-filled circle as the target for nose touch:** A black-filled circle appeared on the white background of the touch-sensitive monitor during preliminary training as the target for the nose touch. Whenever the horse touched the black circle, they were rewarded. Then, the black circle was replaced with the letters of the alphabet for discrimination learning.

**Letters of the alphabet:** Letters were chosen in this study because they can be clearly defined by font and include various features of two-dimensional shapes. We selected five letters, namely O, B, V, Z, and X, in Arial font. The Arial font was chosen because it is one of the most popular “Sans-serif” fonts nowadays. It is also a descendant of Helvetica font, which was used in a study of the perception of letters in chimpanzees [43]. For comparisons across species or in different places, any popular font should be adequate. All stimuli appeared in black against a white screen background. The size of the X stimulus was modified according to the training or test sessions as follows: 5 cm height (X1), 8 cm height (X2), and 15 cm height (X3). When the horses viewed the stimuli from the distance of their eyes to their nose, about 36 cm (see Table 1), the 15-cm-high stimuli had approximately 24 degrees of visual angle, large enough for horses to see them.

**Partition of stimuli:** In the two-choice discrimination task, each stimulus in the pair was presented on one side of the screen. The left and right halves of the screen were separated by a 4-cm thick wooden barrier fixed 8 cm in front of the screen. Thus, the horses could touch either the left or right stimulus, but not both (see Figure 2). Both stimuli disappeared once the horse touched either side of the screen.

**Sound samples:** Four different sounds were automatically played during the programmed experiment. A “bip” sound was played when the shapes appeared on the screen. A “chime” sound was played when the horse’s nose-touch response was correct. A “buzzer” sound was played when the horse’s nose-touch response was incorrect. An “ending” sound was played when the session came to an end. These sounds all had a frequency between 200 and 800 Hz, within the horse’s hearing range of about 55 to 33,500 Hz [53].

### 2.4. Procedures

#### 2.4.1. General Procedure

One horse at a time was separated from the other members of the group and led by the experimenter to the experimental setup in the stall. The experiment started once the horse found four carrot cubes that were pre-placed in the feeder bowl. The experimenter entered the stall to launch the program while the horse ate the carrot cubes. The screen’s background color changed during the session as follows: it was red when the horse arrived in the experimental setup, green during inter-trial intervals, white during the trials, and blue at the end of the session (see Appendix A).

The lead rope was disconnected from the halter once in the testing area. The horse could leave the experiment at any time. Although the rest of the group was not directly in the horse’s field of view in the testing area, a simple rotation of the horse’s head brought the other horses into view, which reduced any possible fear of isolation. In principle, the horses were always together as an OMU (see Appendix A).

A trial proceeded as follows. The participant horse was free to touch the panel, upon which the stimuli automatically appeared with the “bip” sound. When the horse made a correct choice, the feedback sound “chime” was given and a carrot cube was delivered as a reward. If the horse’s choice was incorrect, the feedback sound “buzzer” was given, without any reward. The inter-trial interval (ITI) was set at 5 sec, during which the horse ate any reward they had just received. Automatically after the ITI, the next trial started with the programmed stimulus appearing on the screen. The response time (RT) was defined as the duration from the onset of the stimulus to the computer’s validation of the nose-touch response (See Appendix A).

Each experimental session consisted of 10 trials in general. At the end of a session, there was an “ending sound” and the screen color was changed. The horses could leave the touchscreen testing area at any moment based on their will. Each horse received about 5 sessions (more precisely, 4.97 sessions) per day on average (between 2 and 10 sessions a day), depending on individual circumstances. Sessions could also end depending on the horse’s motivation. They were free to participate in the task and also free to stop; sessions ended because a horse refused to continue in only 23 out of 843 sessions in total (2.7%).

#### 2.4.2. Preliminary Training

**Shaping the nose-touch response:** Training is based on operant conditioning established in the 1950s by C. B. Ferster, B. F. Skinner, and their colleagues [54]. The theoretical framework is called “Experimental analysis of behavior” in which the behavior can be described by 3 terms: discriminative stimulus, operant, and reinforcer. The first step is called “shaping” to shape the desired behavior with positive reinforcement. The positive reinforcements chosen in this study were a portion of edible treats with social praise. Before being introduced to the touchscreen, the horses performed “targeting training” with a black circle target (8 cm diameter) placed on a telescopic probe (30–50 cm long). This preliminary training was first conducted in the field by a skillful horse trainer, BR (Figure 3, Left). The participants were first tested together in the field before being separated from one another. Each nose-touch or mouth-touch response to the target was reinforced with a carrot cube. Each horse received 150 trials over five months, during which the participants gradually became accustomed to the daily routine. Targeting training was then conducted on a magnetic board (Figure 3, Center), followed by training at the entrance of the testing area.

**Nose-touch response on the screen:** The technique called “successive approximation” and “fading” in operant conditioning was introduced in this stage. The horses were individually led to the apparatus. A large black-filled circle (13 cm diameter) was programmed to appear on the center of the screen with the “bip” sound. Whenever the horse touched the target with his/her nose or mouth, the “chime” sound was automatically played, the target disappeared, then the experimenter delivered a carrot cube through the feeder tube. Each horse completed 30 trials of this training phase. The training then continued for 100 trials on average, with the target presented at different places on the touchscreen (Figure 3, Right). This increased the precision of their nose-touch response. Individual adjustments were made during this preliminary training phase: the distance between the entrance barrier and the screen was set at 80 cm for Boneko and Kiki, and 90 cm for Flore, Noven, and Petala.

**Detection of screen touches:** During the preliminary training phase, automated detection of an induced pressure on the touchscreen was used to automatically validate the horses’ nose-touch responses. However, it was not used in the tests that followed because screen sensitivity differences appeared among the participants, which meant that some touches were not detected properly. Therefore, the present study took a combined approach to detection: the horse’s response was visually inspected and manually validated by the experimenter on the keyboard for the following test sessions (see Appendix A for the experimental setup behind the screen).

#### 2.4.3. Discrimination Learning

Through the preliminary training, all horses learned to nose-touch the black circle on the touchscreen. Then, the horses were given a new task in which they had to discriminate shapes. The procedure was again based on positive reinforcement training. During the discrimination training, the same large black-filled circle (13 cm diameter) was always S+ (positive stimulus), and the letter X was always S- (negative stimulus). The X was introduced gradually in increasing order of size (5 cm/8 cm/13 cm, in other words, X1, X2, and X3). We required errorless performances at each size so that horses would be best prepared to succeed in discrimination. Each horse moved on to the next letter size of X only after 100% correct scores on 10 trials in two consecutive sessions.

#### 2.4.4. Baseline Pairs

The test of shape perception consisted of two stages. The first stage was called “Baseline pairs” and the second stage was called “Transfer test”. After completing the training on the black circle vs. the letter X, the horses performed the shape discrimination tests involving the four letters, O, Z, B, and V, in addition to the now-familiar letter X. Initially, the horses were trained to discriminate between O vs. X2 (8 cm), before proceeding to O vs. X3 (15 cm) after reaching the criterion of 100% correct choices. In this sense, O was the “positive” stimulus S+, while X remained the “negative” stimulus S-. The “baseline pair” was always S+ paired with the letter X as S-.

#### 2.4.5. Transfer Test

Once the criterion was reached for the training on the baseline pair O vs. X, X was replaced with one of the other letters: V, B, or Z, paired as a “negative” stimulus, S-. Each pair (O vs. V, O vs. B, and O vs. Z) was presented for three sessions of ten trials each. After finishing the transfer test of O vs. the other stimuli, the next letter “B” became the S+ and the new baseline pair was B vs. X. Thus, the cycle of the baseline pairs and the transfer test was repeated. After training on the baseline pair B vs. X, in the transfer test, B was tested vs. V and vs. Z. The baseline pair then became Z vs. X. The corresponding transfer test presented Z vs. V. At the end, the final pair of V vs. X was tested.

The order of the shape pairs presented was fixed and unchanged for all horses. This was done to detect any individual differences among horses while the conditions were kept constant. It must be noted that the present study did not test the reverse cases involving changing the roles of S+ and S-.

The transfer test proceeded as follows. In the first two sessions, a reminder of the baseline pair was given. This was a short “confirmation session” of 6 trials to check that the horse participant remembered the positive stimulus. If the horse scored at least 4 “correct” out of the 6 trials vs. X, a further transfer test of 10 trials was given. If not, the confirmation session was repeated. As described, X was always used as S- throughout the preliminary training, baseline pairs, and transfer tests.

#### 2.4.6. Special Considerations in the Procedure

**Side bias:** Participants often develop the tendency to always choose either right or left, as both present a 50% chance of getting the reward. To avoid this “position preference”, the order of the trials was randomized by the computer program so that half of the trials in a session presented the positive stimulus, S+, on the right side and the other half on the left side. When the horse reached the criterion in two consecutive sessions with 100% correct choices, this indicated the absence of a side bias at least for the most recent two sessions. This is an important prerequisite to proceed to the transfer test phase.

**Avoiding social cueing (experimenter bias):** The experimenter was always positioned on the right side of the screen when directly validating the horse’s responses in the discrimination training and transfer test phases. She was blind to the stimulus presentation because the computer screen was rotated 180 degrees away from her view to prevent the “Clever Hans phenomenon”. The left-right position of letters was randomized by the computer program. The experimenter had no prior knowledge of which shapes were presented on which sides of the screen. Depending on the correct/incorrect feedback sound of the computer after the horse responded, the experimenter delivered the food reward or not (see Appendix A).

**Special aid for the disabled horse:** The half-blind horse, Flore, was helped during the discrimination learning phase by a slightly modified procedure: the next trial was manually launched only when she turned her head to the right, enabling her to see the whole screen and both shapes with her intact left eye. Otherwise, she rushed to touch any shape on the left side of the screen. After a few sessions, she started to spontaneously tilt her head to see both sides of the screen instead of just the left side. After she learned how to do this, no further special aid was given to her for the subsequent training and test phases.

### 2.5. Data Analysis

**Multidimensional scaling analysis**: Based on error patterns for each pair of letters, the authors created a dissimilarity matrix to analyze perceptual similarity among the letters using multidimensional scaling analysis (MDS) with R [55].

This method yielded spatial representations for the letters as well as weights for each dimension of this representation for each observer horse. A two-dimensional solution was applied. The intraclass correlation coefficient (ICC) was calculated among the data sets to create a confusion matrix for all horses combined.

## 3. Results

### 3.1. Preliminary Training

All five horses successfully learned to discriminate between the black-filled circle vs. X. The number of sessions to reach the learning criterion (100% accuracy in two consecutive sessions) is shown in Figure 4. Individual differences were apparent. Visual inspection indicates an age difference: the three younger horses were better than the two adults at discrimination learning in all four conditions of the preliminary training. The male Boneko, in particular, had difficulty in achieving perfect discrimination learning scores (see Appendix A). The number of sessions needed for the half-blind mare (Flore) to master each stage was longer than the other females. Nevertheless, her performance was better than Boneko’s. Given the small sample size, we conducted no statistical analysis on age and sex effects.

### 3.2. Baseline Pairs

The number of sessions to reach the criterion on each baseline pair is shown in Figure 5. Four out of five horses learned to discriminate between O (S+) and X (S-), B (S+) and X (S-), and Z (S+) and X (S-). The male, Boneko, failed to reach the criterion (see the later section on individual differences). Visual inspection revealed no age effect on the performance. However, as in the preliminary training, the three younger horses were better than the two adults on the first baseline pair (O vs. X).

### 3.3. Transfer Test: Perceptual Similarity of Shapes in Horses

#### 3.3.1. Accuracy of Discriminating Each Pair of Letters

The mean discrimination accuracy of the four horses was good during the transfer test (80.4%, SD 13.7%; see Figure 6). The discrimination was difficult for O vs. B, B vs. Z, and V vs. X. To evaluate consistency among the horses, we calculated the intraclass correlation coefficient (ICC). ICC “agreement” type was ICC (2, 4) = 0.879 (*p* < 0.001) [55], which means that all four horses exhibited similar patterns of perceptual confusion. Combining the four horses’ data, Table 2 shows the dissimilarity matrix for the five letters.

#### 3.3.2. MDS Analysis of the Perceived Similarity of 5 Shapes

To analyze perceptual similarity among these letters, we used accuracy (% correct) as the dissimilarity index. Low accuracy means that the two letters were confusable, and perceived as similar. The dissimilarity matrix (see Table 2) was then used as the input data for a multidimensional scaling analysis (MDS) [55]. Shapes can be described by features such as “Curved”, “Horizontal/Vertical line”, “Orthogonal line”, “Open vs. closed”, and so on [56]. As displayed in Figure 7 (see Appendix A for individual data), the four mares perceived shapes with shared features (O and B of “curved”; V and X of “diagonal”) as close in similarity. Z also shares the horizontal/vertical feature with B and it was perceived as a little closer to the curved shape group (O and B) than to the only diagonal line group (V and X). Features were the determinant of the perceived similarity of the letter shape.

### 3.4. Individual Differences

**Examining the age effect:** The number of sessions required for discrimination learning showed an age-category effect: the three young horses were better than the two adults in all conditions of preliminary training (Figure 3) and the first baseline pairs condition (Figure 4). Despite this apparent trend, however, there was no clear effect of age gradient such as the order of 13-8-5-4-2 years old. No statistical analysis was performed on these data because of the small sample size.

**Response time**: The touch panel system is advantageous for measuring response times as it precisely records the latency from the stimulus onset to the validation of the nose-touch response. The time taken to respond was 3.98 sec on average, ranging from about 2 to 6 s. Response times appeared consistent within each individual (Figure 8). The response time of the male, Boneko, was extremely long in the baseline learning of O vs. X. This is congruent with his difficulty in discrimination learning. Response time might be correlated with the difficulty of the task.

## 4. Discussion

### 4.1. Shape Perception by Horses

Perceived similarity among a set of five letters was measured in five Garrano horses. Following a training period, these horses demonstrated an average discrimination accuracy level of 80.4% (SD 13.7%) during the transfer test. Data analysis using MDS and feature analysis suggests visual shape discrimination by the horses. In particular, the curved shapes O and B were perceived as similar, the straight-line shapes V and X were perceived as similar, and the letter Z was located in between, connecting the two groups. This might be due to the “horizontal/vertical” feature that was shared between the letters Z and B. The present study is consistent with the visual perception of shapes being somehow common to diverse species living in different niches, including pigeons (avian, [44,45]), chimpanzees (terrestrial-arboreal, [43]), dolphins (aquatic, [46]), and horses and humans (terrestrial: horses [47]; humans [42]).

This provides the further possibility of using letters of the alphabet for communication. Horses are known to do symbolic communication [48] while chimpanzees learn to use the alphabet to identify each individual [43]. Accordingly, letters of the alphabet can be used for symbolic communication in horses. Hopefully, this direction of horse cognition studies may lead us to a new study of zoo semiotics [49].

Further tests on shape perception may shed light on the features [56] influencing shape perception in humans and nonhumans. For example, it might be important to examine how shape discrimination would be affected if a motion component were incorporated into the testing. This is the merit of using the touchscreen system in which the experimenter can easily manipulate the shape and motion on the screen. Another interesting topic is anisotropy, which means the sensitivity difference depending on the spatial orientation [57]. As horses and humans are terrestrial, they might be more sensitive to the gravity control of horizontal and vertical frames than avian, aquatic, or arboreal species who more often change their body orientation in three-dimensional space. Response times might be an important complementary index to accuracy in perceptual and cognitive tests. However, for future studies, we suggest several improvements to the apparatus (see Appendix A.

### 4.2. Cognitive Task in the Social Environment

The present study demonstrated that the computer-controlled touchscreen system worked well to test visual discrimination in Garrano horses similar to other horses [47,58]. The touchscreen system allowed precise control of the stimulus and eliminated potential social cueing of the Clever Hans type. In addition to its value for psychophysical testing, the system allowed horses to be tested in their social environment. They were free to engage in the task, or possibly even more importantly, free to quit the session and return to the group at any time. The target group is also unique because of the one-male unit (OMU) of five horses. It simulated the social group of Garrano horses in the wild [5,6,7,8,9,10,11,12]. There was a stallion and two adult females and two offspring. The stallion, Boneko, was castrated so there was a lack of testosterone which may affect the social structure of the one-male unit. This is an initial constraint to keep the social group in captivity. The present study successfully introduced the touchscreen study of cognition to this socially organized group. Based on their free will, the horses started the discrete trial and stopped it by following the ITI signals. This kind of freedom may be important in terms of animal welfare.

### 4.3. Individual Differences among Horses

The five participant horses were not homogenous. Nonetheless, individual differences in test performances are expected and may be attributed to various factors including age and sex. A possible age effect was observed, as the three young horses showed better discrimination performances than the two adult horses. A sex difference in horses (females outperform males) has been reported in operant target learning [59]. With only one male horse, the present data are insufficient to draw any statistically valid conclusion in this regard. However, we tentatively suggest that the male’s difficulties in discrimination learning might be related to his social role as a stallion and protector of his family group (the OMU), reducing his attention to the learning task and the apparatus. Similar research with bigger samples (or different groups) could shed further light on the possible sex differences in horse attention and learning.

### 4.4. Future Perspectives

The horse’s visual acuity has been reported as 23.3 cycles per degree in spatial frequency discrimination [41]. The present apparatus can contribute to further investigations of horse visual perception. Studies of the visual system will benefit our understanding of equine perception, cognition, and their links with learning and training. The touchscreen system used here opens the window to a range of questions about horse cognition. Multiple future directions for studies using this system in horses can be imagined. Comparisons among all 26 letters of the alphabet is one option. Motion can be introduced to the letter perception; to this end, the computer-controlled touchscreen system is convenient for manipulating images on the screen.

Lansade et al. (2018) successfully showed that horses can identify and memorize human faces [58]. How about the identification of horses by horses? Using the same two-choice discrimination as in this pilot study, letters could easily be replaced with photos of horses, for example, to test discrimination of horse portraits that belong to the participant’s OMU or herd (consisting of several OMUs and all-male units) compared with unknown horse portraits.

Finally, the touchscreen system can be easily applied to other species. The extant horse family consists of only one genus, named *Equus.* The donkey (*Equus asinus*) is a sister species to the modern horse (*Equus caballus*). These two species have both evolved and diverged from their common ancestor. Although differences in visual perception and cognition within the Equidae family appear likely, donkeys have been neglected in psychological studies. They might be smarter than many people believe. The touchscreen protocol used here could be extended to a comparison of visual perception and cognition between the Garrano horse and the Miranda donkey, another endangered local species from Portugal.

## 5. Conclusions

Recent advances in field research have revealed several aspects of the social life of feral Garrano horses in northern Portugal. However, very little is known about the mind of horses. The present study applied a touchscreen system to study their visual perception. All five horses (13, 8, 5, 4, and 2 years old) successfully acquired the skill of nose touch, and all except the male learned to discriminate five letters of the alphabet, namely O, B, Z, V, and X. The confusion matrix of letter pairs was used to conduct an MDS analysis. The results showed perceptual similarities among letters with curvatures compared with those with straight lines. Shape perception in horses shares commonalities with that of humans and other animals (chimpanzees, pigeons, and dolphins) living in different niches. Because of the nature of the automated testing system, the touchscreen discrimination was free from the “Clever Hans phenomenon”.

The present study also showed a new testing setup for visual cognition in horses. The participant horses lived together permanently in a semi-natural group in an enriched environment. This social situation simulated a natural “one-male unit (OMU)” of horses in the wild, consisting of one stallion and multiple females and their dependents. The horses were not kept in the isolated stables but free on the outside. We conducted cognitive studies of horses on-site in their familiar habitat in northern Portugal. A subject horse was guided to the touchscreen. Then, the lead rope was disconnected from the halter once in the testing area. The horses were free to engage in the tests and also free to leave and return to the social group. Thus, the automated touchscreen system can allow greater freedom for group-living horses in an environment close to their natural habitat.

## Figures and Tables

**Figure 1 animals-12-03514-f001:**
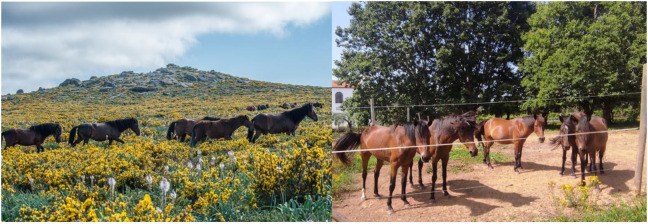
(**Left**): Garrano horses in the wild in Serra d’Arga. The feral horses live in a family group called the one-male unit (OMU), consisting of one adult male and multiple females and their dependents. (**Right**): A group of five captive participant horses simulating a natural family group (OMU). Their enclosure is located within walking distance (about 9 km) of their natural habitat, in Lanheses, Viana do Castelo. (Photos by Barbara Ryckewaert).

**Figure 2 animals-12-03514-f002:**
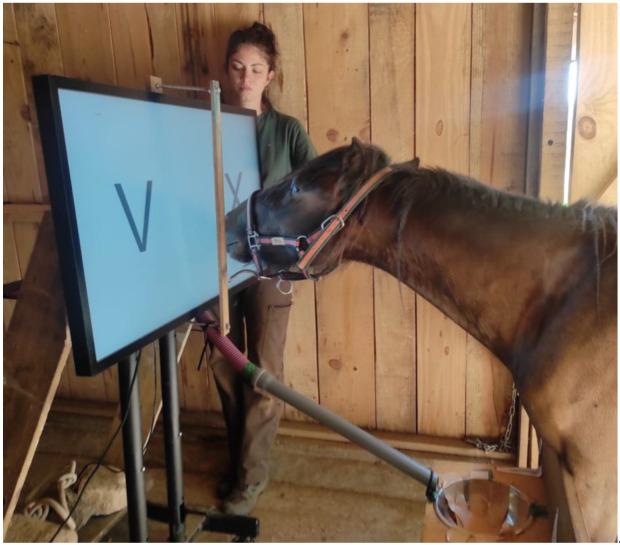
Experimental setup. A touchscreen (touch panel and display) displays the stimuli and automatically detects the nose/mouth touch by the horses. The touchscreen was divided into two halves. The left-right sides of the screen were separated by a 4-cm thick wooden barrier fixed 8 cm in front of the screen. Thus, the horse was allowed to touch either the left or right stimulus, but not both. Here, the horse is about to touch the letter “X” (S-) on the right half of the touch panel; the letter “V” (S+) is on the left half of the screen. Therefore, this is an example of an error trial. Horses were free to participate or quit the experiment at any time: the lead rope was disconnected from the halter once in the testing area. However, entering the stall was prevented by a 70-cm high wooden barrier (see above); the horse responded by extending his/her neck over the barrier and his/her head toward the screen. The reward was delivered manually through a tube and landed in a bowl below the touch panel in front of the horse (see Appendix A).

**Figure 3 animals-12-03514-f003:**
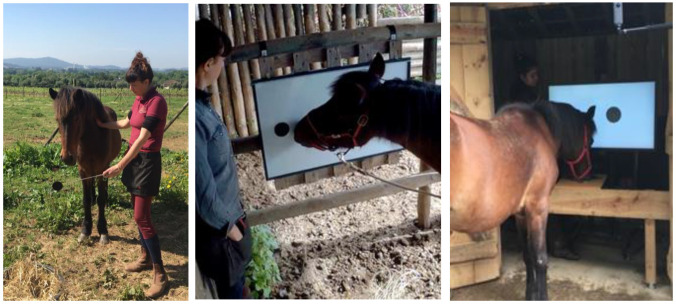
Targeting training with the horse Boneko. (**Left**): Presentation of a black circle target on a telescopic probe in the field. (**Center**): Presentation of the black circle target on a magnetic board. (**Right**): Targeting training performed on the computer-controlled touchscreen with the black circle target. During the preliminary training, the horses gradually habituated to individual testing, away from the rest of the OMU. Without the lead rope connected to the halter, the horse was free to approach the apparatus (Photos by Barbara Ryckewaert).

**Figure 4 animals-12-03514-f004:**
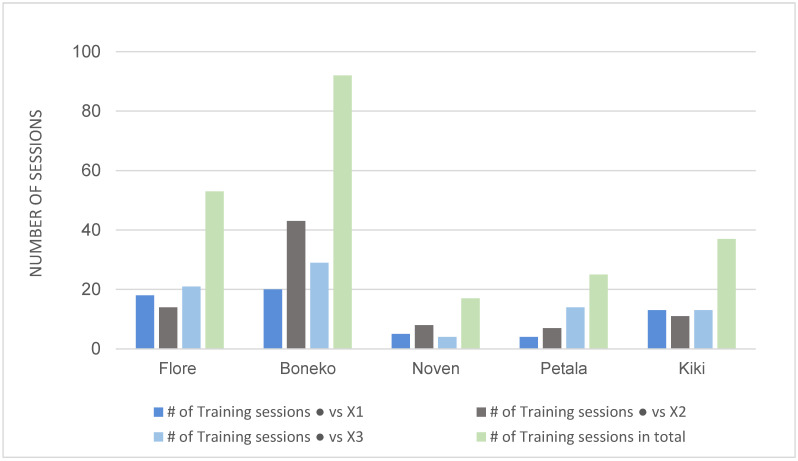
The number of discrimination learning sessions for each horse. Training proceeded in three stages of changing the size of the S- (X1: 5 cm, X2: 8 cm, and X3: 15 cm, see Methods 2-4-3). The two sessions with 100% accuracy were included in the total number of sessions. The *X*-axis shows the horses in order of age (13, 8, 5, 4, and 2 years old). Special aid was given to the disabled horse, Flore, in this preliminary training (see Appendix A for the raw data).

**Figure 5 animals-12-03514-f005:**
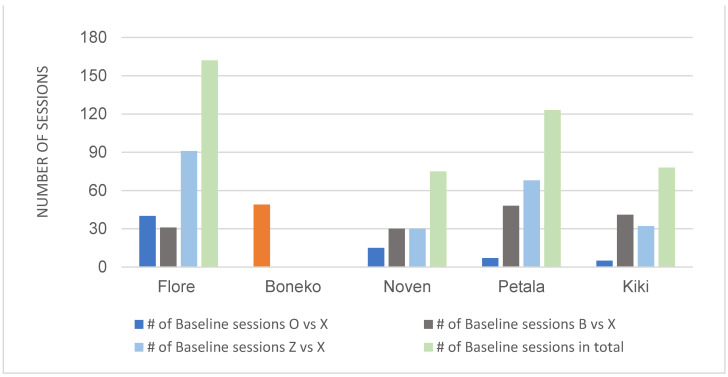
The number of baseline pair discrimination sessions for each horse (see Methods 2-4-4). The male, Boneko (in orange), did not reach the criterion so received no further training. The *X*-axis shows the horses in decreasing order of age (13, 8, 5, 4, and 2 years old).

**Figure 6 animals-12-03514-f006:**
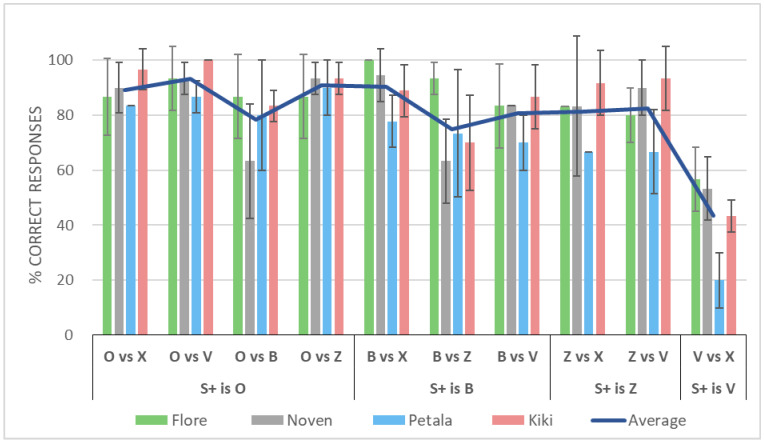
Accuracy (% correct) of discrimination between each pair of letters in the transfer test (See Methods 2-4-5). Discrimination was difficult for pairs O and B, B and Z, and V and X. Only the four mares completed this phase.

**Figure 7 animals-12-03514-f007:**
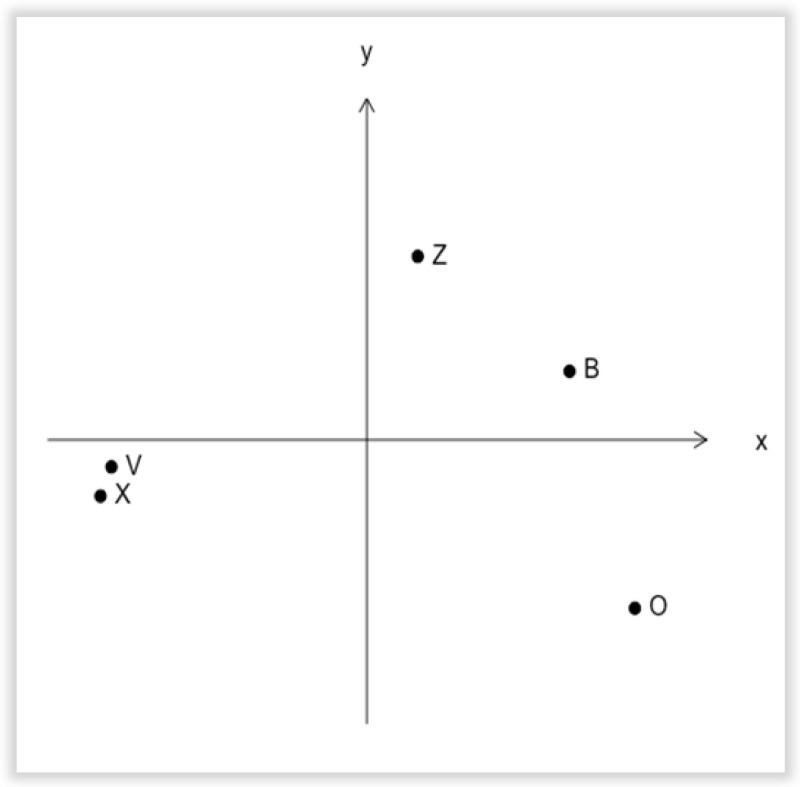
Perceived similarity among five letters. A two-dimensional MDS solution was applied to the dissimilarity matrix (see Table 2) calculated from the combined data of the four mares. (Appendix A displays the individual two-dimension solution MDS for each mare. They showed the same perceived similarity).

**Figure 8 animals-12-03514-f008:**
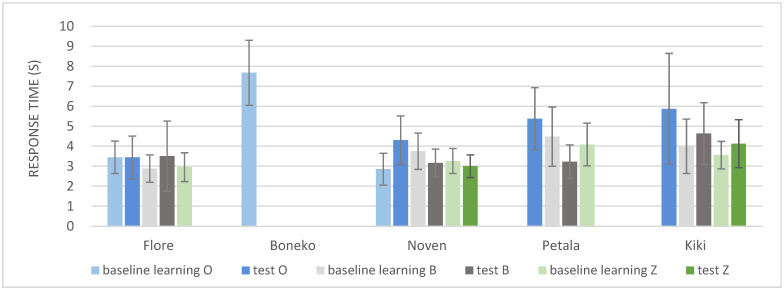
Response times of each horse participant in each task. Some baseline and test phase data are missing because of a technical problem. Only the baseline learning data of the letter O vs. X is indicated for Boneko because he did not achieve the success criterion in this phase to move on to the next phases. The baseline learning data of response time for O is missing for Petala and Kiki because of a technical problem with the computer, as is the test data of the Z letter for Flore and Kiki. However, the consistency of response time within each individual across the phases appears clear. Boneko’s response time was considerably longer than the other horses. The long response time might reflect the difficulty of discrimination learning.

**Table 1 animals-12-03514-t001:** Details of the participant horses. Name and gender of each horse participant. Additional information on the link between each horse and their visual status is given in the Notes column. The Distance eyes/nose for each horse is defined by the distance from the center of the eyeball to the nose tip in the side view. The distance can impact the discrimination study using the touchscreen. The measure of the withers height is indicated for each participant. It was important to choose the distance between the entrance barrier and the touchscreen.

Name	Age	Sex	Note	Distance of Eyes and Nose (cm)	Withers Height (cm)
Flore	13	female	one-eye blind	37	142
Boneko	8	male	castrated	33	124
Noven	5	female	daughter of Flore	36	128
Petala	4	female	daughter of Flore	36	134
Kiki	2	female	daughter of Noven	37	120

**Table 2 animals-12-03514-t002:** Dissimilarity matrix of the 5 letters for 4 horses. The value is the accuracy (% correct) in the discrimination task in the transfer tests. S+ means the positive stimulus while S- means the negative stimulus. Touching S+ was rewarded in each pair. In the present study, the upper right of the matrix is the real result of discrimination learning. The lower left of the matrix is the mirror image to calculate the perceived dissimilarity (perceived distance) of two letters. The present study did not test the reversed situation of changing the role of S+ and S- in each pair. Letter O was always S+, letter X was always S-, and letters B, Z, and V were in between.

	S-					
S+	O	B	Z	V	X	Average
**O**	-	78.3	90.8	90.3	89.2	87.9
**B**	78.3	-	75.0	80.8	90.3	81.1
**Z**	90.8	75.0	-	82.5	80.5	82.2
**V**	93.3	80.8	82.5	-	43.3	75.0
**X**	89.2	90.3	80.5	43.3	-	75.8
**Average**	87.9	81.1	82.2	75.0	75.8	**80.4**

## Data Availability

The raw data are available on the following site in Figshare Dataset: Schubert, Claralynn; Matsuzawa, Tetsuro (2022): Data of the Horse touch panel study published in *Animals* in 2022. https://doi.org/10.6084/m9.figshare.21694670.v1.

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
