# Peer review of "Garrano Horses Perceive Letters of the Alphabet on a Touchscreen System: A Pilot Study"

_animals, 2022, doi:10.3390/ani12243514_

Round 1

Reviewer 1 Report

Boneko (gelding) - question whether lack of testosterone may affect social structure -- may want to include discussion of social behavioural deviation(s) from a true OMU. 

Figure 9, not clear as to whether Patala  and Kiki did baseline learning since response time O conditioning not apparent in bar graphic and not clear as to why authors chose to not show some results here.  

Position of the 'rewarder' might be better behind the screen from pony rather than to one side since it effects the symmetry of the exp. The idea is presented in the supplementary material (A7) by automating the reward.

I would be curious as to how shape discrimination would would be affected if a motion component were incorporated in the testing.

Reviewer 2 Report

This is an interesting study looking at the ability of ponies to discriminate between letters of the alphabet. While the idea itself is interesting, the background information does not provide any justification for the study or relevance to the objectives. There is a fair bit of research done regarding object recognition in horses which is not cited.

Similarly, the discussion does not link to previous studies or form a concrete association with horse cognition. Again, there is much literature regarding horse cognition and I would specifically draw the authors’ attention to brubaker and udell’s recent review of horse cognition (Cognition and learning in horses (Equus caballus): What we know and why we should ask more. Beh Proc 2016, 126:121-131).

Finally this research reports on findings from 5 ponies (actually only 4 since one failed to learn the task) and does not report any statistical analyses (rightly) and it should be made clear that this is a preliminary or pilot study, particularly in the title.

 L40 – keywords should not already appear in the title

 L45-50 – it is unclear what the significance of these sentences are other than to introduce the Garrano breed. However, no real information is provided as written. 8 references are cited without any clarification of the Garrano’s social life. However this is explained in L78-84. Perhaps the first paragraph is not necessary and would be better served to introduce the topic of object discrimination.

 L57 – I would point the authors to rorvang’s recent paper which appears in your reference list (37) but is not actually cited in the paper

  L94 – while this may not be relevant to animals, the term disabled is preferred over the term handicapped for humans.

 Introduction – does not clearly provide background information that logically leads the reader to the objectives of the study.

 L103 – the Garranos were described as ponies on the previous line so should be referred to as ponies throughout the manuscript

 L147 – here the barrier is said to be 70cm high while in the caption for Figure 2 it is said to be 80 or 90 cm high.

 L148 – as horses are sentient animals they should not be referred to as “it”. please change here and throughout the manuscript.

 L156 – did the pony receive the apple through the dispenser or directly from the experimenter?

 L182-183 – here in the methodology mention is made of other comparative cognition studies. This would be better placed in the introduction. Same with the topic of shape perception. Since this is your research question, the introduction should cover the background knowledge of this topic. Same with the information on shape features on L193-198.

 L220 – I suspect that by reins you actually mean the lead rope (just terminology. Reins are attached to a bridle while a lead rope is attached to a halter)

 L229 – here you mention the ITI was 5s while on L154 you stated that the inter-trial intervals depended on how long the horse spent chewing the carrot cubes.

 L350 – please expand the caption for Table 2 to fully explain what it depicts.

 L360 – I am not sure what is meant by “age-graded effect”

 L505 – Reference 34 appears after 35 and 36.

 Discussion – I would caution the authors to discuss the results guardedly. Given the small and non-homogenous sample size, it is difficult to make any conclusions

Missing key references on object discrimination in horses (see work by Hanggi, Mejdell, Winther Christensen and others)

I am not convinced that the social organization of horses is clearly linked to this study’s objectives. Also while testing within the ponies regular habitat may well be useful, this was not explicitly tested so no conclusions can be drawn about it. The relationship to welfare is tenuous at best.

 Careful throughout the manuscript as you often switch between past and present verb tense.

Reviewer 3 Report

The introduction should include more information on horse visual perception, and why visual discrimination knowledge is useful for human caretakers (as is implied, but not described or justified). More information in the methods on operant conditioning should be included; such as, what role the sounds play as conditioned cues. Many of the figures provided are not intuitive or helpful, suggest removal of Table 2 and Fig 8. The discussion did not provide a thorough interpretation of results (such as, response time, similarity perception, sex differences), with too much emphasis on the potential application of a touchscreen tasks with equine (which is already established in the literature). The authors should expand on the concept they briefly touched on in lines 465-467: why is detail visual perception important or NOT important for this grazing herbivore? Why would individual identification using vision, rather than olfactory or auditory cues, be important for this species?

Simple summary: Should remove any reference to animal welfare as this study does not evaluate welfare nor does it have a clear objective related to welfare.

Introduction

There are far too many rhetorical questions in the introduction. It is recommended that the authors removed these (lines 45, 77, 86, 475, 505)

Line 64: Parenthesis bracket missing

Lines 89-98: This is information for the methods, not the introduction. The authors need to include some hypotheses or predictions. What are the objectives?

Methods

Line 131: What are they fed? What are the housing and management styles? Size of pen? What humans interact with these 5 ponies?

Line 175: It is recommended that the authors set up this section as a step progression (each goal described, as the experiment reaches the more complicated task)

Line 180: Which alphabet? Arabic?

Line 185: Where is the citation for this font being popular? Why does this matter for horses?

Table 1. This caption is not sufficient. More information is needed. It is not clear what is being presenting in this table.

Figure 4. Capitalize y-axis title

Table 3. How did the authors come up with these attributes ? Why did they not include more attributes? A more thorough justification for these particular symbols is needed.

Discussion

Lines 471-480: This section should be removed or enhanced. There is no evidence provided that letter discrimination or touchscreen tasks can elicit information about animal welfare. This is speculation with no scientific sources.

Round 2

Reviewer 2 Report

Thank you to the authors for a thorough revision of their manuscript. It flows much better now and tells a good story.

Thank you for not referring to the horse as "it". However there was one instance missed on L227. It is grammatically correct to use the plural they/their even when only referring to one animal. So this line could read "Whenever the horse touched the black circle they were rewarded."

Author Response

Thank you very much.

Reviewer 3 Report

All concerns and suggested edits have been addressed by the authors.

Author Response

Thank you very much.
